# Role of Disulfide Bonds and Sulfhydryl Blocked by *N*-Ethylmaleimide on the Properties of Different Protein-Stabilized Emulsions

**DOI:** 10.3390/foods10123079

**Published:** 2021-12-10

**Authors:** Mangang Wu, Zhikun Li, Ranran Wei, Yi Luan, Juan Hu, Qingling Wang, Rui Liu, Qingfeng Ge, Hai Yu

**Affiliations:** 1College of Food Science and Engineering, Yangzhou University, Yangzhou 225127, China; lizhikunlzk@163.com (Z.L.); wrr_1221@163.com (R.W.); LY19980000@163.com (Y.L.); hj1148019303@163.com (J.H.); qlw@yzu.edu.cn (Q.W.); ruiliu@yzu.edu.cn (R.L.); qfge@yzu.edu.cn (Q.G.); yuhai@yzu.edu.cn (H.Y.); 2Industrial Engineering Center for Huaiyang Cuisine of Jiangsu Province, Yangzhou University, Yangzhou 225127, China

**Keywords:** myofibrillar protein, sulfhydryl-blocking agent, disulfide bond, protein-stabilized emulsions, interface protein membrane

## Abstract

To investigate the role of sulfhydryl groups and disulfide bonds in different protein-stabilized emulsions, *N*-ethylmaleimide (NEM) was used as a sulfhydryl-blocking agent added in the emulsion. The addition of NEM to block the sulfhydryl groups resulted in a reduction in disulfide bond formation, which enabled the internal structure of the protein molecule to be destroyed, and then decreased the restriction of protein membrane on the oil droplets. Furthermore, with the NEM content increasing in the emulsion, a reduction in the protein emulsifying activity and emulsion stability also occurred. At the same time, the intermolecular interaction of the protein on the oil droplet interface membrane was destroyed, and the emulsion droplet size increased with the NEM content in the emulsion. Although NEM blocking sulfhydryl groups from forming disulfide bonds has similar effects on three types of protein emulsion, the degree of myofibrillar protein (MP), egg-white protein isolate (EPI), and soybean protein isolate (SPI) used as emulsifiers had a subtle difference.

## 1. Introduction

Proteins are often used as emulsifiers for their amphiphilic nature and film-forming abilities [1]. The emulsion composed of oil and protein has been widely used as fat substitutes in meat products. When protein molecules enter the oil surface in the emulsion, the molecules continue to be adsorbed on the oil–water interface. Consequently, protein conformation changes occur, leading to molecule unfolding and exposure of internal hydrophobic groups, the process of which is called “interfacial degeneration” [2]. The stability of a protein emulsion depends mainly on the ability of “interface degeneration”, which can be attributed to the ability to form a viscoelastic interface membrane. In addition, the formation of interfacial membrane relies on non-covalent interactions, hydrogen bonds, hydrophobic and electrostatic interactions, and covalent interactions of adsorbed protein molecules, such as disulfide bonds [3]. Dickinson and Matsumura gave the first direct evidence that similar conformational changes and the associated intermolecular disulfide bond formation occurs during the adsorption of *β*-lactoglobulin at the oil–water interface [4]. In our previous study, we also investigated the effects of disulfide bond formation between protein-coated oil droplets and the surrounding protein matrix, and its contribution to the rheological properties of myofibrillar protein (MP)–emulsion composite gels [5].

Disulfide bonds are formed via the oxidation of thiols on two cysteine residues of amino acid side chains on a known protein polypeptide [6]. Disulfide bonds are a covalent bond, but they are not very solid. Reduction reactions and a transfer to sulfhydryl fracture occur easily, while sulfhydryl groups can be re-oxidized to form a disulfide bond. In conclusion, sulfhydryl groups and disulfide bonds can be transformed into each other at a certain state. In other words, the content of disulfide bonds is dynamic and only has relative stability. It has been reported that disulfide bonds maintain many of the protein-specific structures and functions [7].

Jones found that, at the early stage of oil–water interface formation, free myosin molecules form a monolayer at the oil–water interface with a relatively intact surface monomer, where the heavy chains face the oil phase and the light chains face the aqueous phase [8]. Other protein molecules achieve protein–protein interactions primarily through hydrophobic interactions, covalent bonds, and hydrogen bonds and eventually form a semi-rigid membrane as other myofibrillar proteins become denser. In some cases, changes in the contents of sulfhydryl and disulfide bonds have a most significant influence on the molecular structure of soybean oil droplets [9,10]. In addition, several studies mentioned adding various blocking agents for further exploration, for instance, dithiothreitol (DTT), *β*-mercaptoethanol, *N*-ethylmaleimide (NEM), and diamide. Previous reports have indicated that DTT, as a small organic reducing agent, could prevent the formation of disulfide bonds when added to myosin [11]. Although this could not change the heat transfer temperature of myosin, the initial temperature for the formation of a myosin gel increased, thus reducing the gel strength and showing that the disulfide bond plays an important role in the aggregation and formation of the gel network.

In this study, different protein emulsions of myofibrillar protein (MP, as a representation of animal meat protein), egg-white protein isolate (EPI, as a representation of animal non-meat protein), and soybean protein isolate (SPI, as a representation of vegetable protein) were studied. NEM was added as blocking agent, blocking sulfhydryl group transfer to disulfide bonds, to investigate the effect of disulfide bonds on the emusification abilities, apparent viscosity, surface hydrophobicity, and other properties of emulsions stabilized by different protein.

## 2. Materials and Methods

### 2.1. Muscle Samples and Other Materials

Fresh pork center loin muscles (pH 5.6–5.9) were purchased from a local market (24–48 h post-mortem). The meat was transported on ice to the university’s processing facility. Visible external fat and connective tissue were trimmed. The remaining meat was diced into approximately 1 × 1 × 1 cm^3^ cubes, and then, samples were placed in plastic vacuum package bags, evacuated, and stored at −70 °C until further use.

Olive oil was purchased from Sinopharm Chemical Reagent Co., Ltd. (Beijing, China). Egg-white protein isolate (EPI) was supplied by Qianyu Co. (Zhejiang, China). Soy protein isolate (SPI) was obtained from Gushen Chemical Co. (Shandong, China).

### 2.2. Preparation of Myofibrillar Protein

MP was extracted from thawed pork muscle samples at 2–4 °C using an isolation buffer (pH 7.0) consisting of 0.1 M NaCl, 10 mM sodium phosphate, 2 mM MgCl_2_, and 1 mM EGTA, as previously described [12]. Isolated MP pellets were washed twice each by suspension in four volumes of 0.1 M NaCl, adjusting the pH of the suspension to 6.2, followed by centrifugation at 2000× *g*. Purified MP pellets were stored on crushed ice and used within two days after isolation. The protein concentration was determined via the Biuret method using bovine serum albumin as the standard [13].

### 2.3. Preparation of Emulsions

Emulsions stabilized via proteins were prepared by mixing 4 g of olive oil with 16 g of diluted MP or non-meat proteins (SPI or EPI) solution, in which the protein concentration was 1% (*w*/*w*), in 0.6 M NaCl and 50 mM sodium phosphate, with a pH of 6.2. This mixture was homogenized at 12,000 rpm for 1 min using an Ultra-Turrax homogeniser (IKA T18 Basic; IKA, Staufen, Germany) to generate pre-emulsion. Afterwards, each protein-stabilized pre-emulsion was mixed with 0, 1, 5, or 10 mM NEM (*w*/*w*), followed by a moderate stirring processing at 25 °C for 30 min at low speed (200 rpm) to ensure complete reaction between NEM with SH. We defined the sample that formed using only high-speed homogenization as the pre-emulsification group. The sample that formed using both high-speed homogenization and low-speed stirring with 0 NEM was defined as the control group. NEM-treated samples were defined as the 1 mM, 5 mM, and 10 mM groups.

### 2.4. Total and Reactive Sulfhydryl Groups

Total and reactive sulfhydryl groups (SH) were determined according to the method by Yongsawatdigul and Park, with several modifications [14]. To 0.1 mL of emulsions, with a concentration of protein of 10 mg/mL, 1 mL of a buffer containing 0.6 M NaCl, 8 M urea, 20 mM sodium phosphate buffer, 10 mM ethylenediaminetetraacetic acid (pH 7.0) was added. To this mixture, 0.4 mL of 10 mM DTNB (5,5′-dithiobis (2-nitrobenzoic acid)) (pH 7.0) was added [15]. The mixture was incubated at 37 °C for 30 min. The absorbance was measured at 412 nm to calculate the total SH groups. Furthermore, the reactive SH groups were conducted using the same method in the absence of urea. The content of sulfhydryl groups was calculated with Equation (1).
(1)$$C_{0}=\left(\frac{A}{e}\right)\times D\times 10^{6}$$
where *C*_0_ represents the concentration (mol/1000 g protein) of sulfhydryl groups, *A* represents the initial absorbance of the mixture (taken at 412 nm), *e* represents the extinction coefficient of 13,600 M^−1^ cm^−1^, and *D* represents the dilution factor.

### 2.5. Surface Hydrophobicity

The surface hydrophobicity of different types of protein-stabilized emulsions was determined using 8-Anilino-1-naphthalenesulfonic acid (ANS) as a fluorescent probe. The emulsions were centrifuged (15,000× *g*, 15 min), and the supernatant was reserved. The protein concentration of the supernatant was determined using the Lorry method [16]. The samples were diluted with 10 mM phosphate buffer (pH 7.0) to yield final concentrations of 0.10–0.20 mg/mL. To 4 mL of different concentrations of diluted samples, 20 μL of the ANS solution (8 mM in phosphate buffer, pH 7.0) was added immediately before reaction in the dark for 1 min. The fluorescence intensity (FI) was measured at 390 nm (excitation) and 470 nm (emission) using a fluorescence spectrophotometer (Cleande, Inc., Jiangsu, China). The slope of the linear curve was obtained using the protein concentration as the abscissa, and the fluorescence intensity as the vertical axis was taken as an index of surface hydrophobicity (H_0_) of the protein supernatant.

### 2.6. Emulsifying Properties

The emulsifying activity index (EAI) and the emulsion stability index (ESI) were determined according to the method of Pearce and Kinsella [17], and Guo and Mu [18]. For the EAI measurement, a volume of 20 μL of freshly prepared emulsions was taken from the bottom of the homogenized emulsions immediately (0 min) after homogenization, and then, 5 mL 0.1% (*w*/*v*) SDS solution was added. The absorbance of the emulsion at 500 nm was immediately measured. The EAI values were calculated using Equation (2).
(2)$$\mathrm{EAI}\left(m^{2}\cdot g^{-1}\right)=\frac{4.606\times A_{0}}{C\times \left(1-\varphi \right)}\times 10^{-4}\times D$$
where *A_0_* represents the absorbance of the emulsions (taken at 0 min), *C* represents the protein concentration (g/mL) before emulsification, *φ* represents the oil volume fraction (*v*/*v*) of the emulsion and *D* represents the dilution factor.

For the ESI measurement, the aliquots were obtained from the bottom of the homogenized emulsions exactly at 10 min, 30 min, 60 min, 120 min, and 180 min after homogenization; then, SDS was added and measured for absorbance at 500 nm, as described above. The ESI values were calculated using Equation (3).
(3)$$\mathrm{ESI}=\frac{A_{t}}{A_{0}}$$
where *A*_0_ and *A_t_* represent the absorbance of the emulsions taken at 0 min and t min, respectively.

### 2.7. Viscosity of Emulsions

The apparent viscosity of emulsions was determined via flow measurements using shear rate assays. Measurements were performed at 25 °C in a Rheometer KINEXUS Pro (Malvern Instrument, Inc., Worcestershire, UK) equipped with PU40 probe. The sampling parameters were a shear rate range of 0.01–1 s^−1^ and a gap of 1.0 mm [19].

### 2.8. Light Microscopy

Light microscopy was conducted according to the method of Wu, Xiong, and Chen with some modifications [5]. A drop of freshly prepared protein emulsion was taken on a clean glass slide. Measurements were performed in an inverted microscope XDS-600C (Caikon, Inc., Shanghai, China), which was connected to a computer, to observe, record, and compare the microstructure of the emulsion.

### 2.9. Statistical Analysis

The reported values are mean and standard deviations for at least three trials. In each repeated trial, triplicate samples were analyzed. An analysis of variance (ANOVA) was performed using the SPSS program (SPSS Statistical Software, Inc., Chicago, IL, USA) to detect significant treatment effects. Significant differences (*p* < 0.05) between the means were identified using Duncan’s multiple range tests.

## 3. Results and Discussion

### 3.1. Sulfhydryl and Disulfide Bond Contents in the Three Protein Emulsion

Sulfhydryl groups are related to the formation of protein aggregates and the degree of protein denaturation [20]. As shown in Figure 1, the total and reactive sulfhydryl groups of the three types of protein emulsions in pre-emulsification group were significantly lower than those of the control (*p* < 0.05). Compared with the former, this is because the control group continuously received a low-speed stir for 30 min, which is equal to further emulsification. The pre-emulsification group emulsions were formed only by high-speed homogenization (12,000 rpm) for 1 min. Once the proteins are adsorbed on the oil surface with homogenization, their conformation may change considerably at hydrophobic surfaces. Such conformational changes can be regarded as a form of interfacial denaturation of the protein [21]. With the further emulsification (we identified it as the control group), we supposed that the energy input (stirring 30 min) was enough to produce abundant reactive sulfhydryl groups with the continuing conformation change. This suggests that some changes occur in the structure of the proteins with continuous emulsification, and these results are similar to the results of Li et al. [22]. In contrast, with the addition of NEM in the protein emulsion, sulfhydryl (SH) groups reacted to form NEM-SH, which made the sulfhydryl not participate in the subsequent reactions. Therefore, NEM affected the content of total and reactive sulfhydryl groups and disulfide bonds. As shown in Figure 1, with the increase in NEM content, total and reactive sulfhydryl groups decreased continuously, especially when the content of NEM increased from 1 mM to 5 mM. Indeed, an analysis of the reactive SH groups showed that 17.41%, 19.42%, and 27.16% sulfhydryls in MP, EPI, and SPI emulsions were lost upon treatment with 1 mM NEM compared with the control, while 68.73%, 44.65%, and 68.14% sulfhydryls in MP, EPI, and SPI emulsions were lost upon treatment with 5 mM NEM (Figure 1B). Reactive SH decreased much more with 5 mM NEM treated than that treated with 1 mM NEM. The total SH group change was consistent with the results of reactive SH. Tong et al. reported that, when 0 mM NEM was added to whey protein emulsions (10% whey protein, 40% salmon oil, and 4% Tween 20, *v*/*v*), reactive SH groups were less than the group with 0.5 mM NEM (*p* > 0.05), while slightly higher than the group with 1 mM NEM. When the NEM content increased from 5 mM to 10 mM, the reduction in free SH groups was most pronounced. Our further analysis showed a significant positive correlation (*p* < 0.05) among the total or reactive SH groups with a NEM content similar to the results of Tong et al. [23].

It can be seen from Figure 1 that emulsions stabilized by different types of protein differed significantly. Myofibrillar protein (MP) is a type of meat protein, which is mainly composed of myosin and actin. Myosin contains 42 sulfhydryl groups, one third of which is embedded in the head of the molecule [24]. Therefore, sulfhydryl groups and disulfide bonds are abundant in MP emulsion. As a non-meat protein, egg-white protein isolate (EPI) mainly consists of ovalbumin, ovotransferrin, ovomucin, and lysozyme, which has good gelling, emulsifying, and foaming properties [25]. An egg albumin molecule contains four embedded free sulfhydryl groups and one disulfide bond [26]. As a result, MP had at least 10 times more SH groups than EPI. There may be less disulfide bonds in the EPI emulsion. Soy protein isolate (SPI), a type of non-meat vegetable protein, is mainly composed of two proteins: glycinin and *β*-conglycinin. Glycinin is a heteromeric hexamer protein with a molecular weight of up to 300–380 kDa and consists of five subunits, namely A_1_aB_1_b, A_1_bB_2_, and A_2_B_1_a of group I and A_5_A_4_B_3_ and A_3_B_4_ of group II. In addition to the A_3_B_4_ subunit, each acidic subunit is linked to a basic subunit via a disulfide bond and ultimately constitutes a glycinin monomer. It has been reported that disulfide bonds contribute to the stability of the glycinin structure [27,28]. *β*-conglycinin is a trimeric protein with a molecular weight of up to 150–200 kDa and consists of three subunits: *α*, *α′*, and *β* [29]. Compared with glycinin, *β*-conglycinin has less disulfide bonds, mainly in the form of linked peptides, which are present in the interior of protein particles [9]. The differences in types and structures of the three proteins caused different contents of sulfhydryl and disulfide in the formed protein emulsions. The content of sulfhydryl groups and disulfide bonds decreased with the increase in treatment level (0 mM, 1 mM, 5 mM, and 10 mM NEM) in each experimental group of the three protein emulsions. Combining the three emulsions for comparison, the content of reactive sulfhydryl and total sulfhydryls in the MP emulsions was the highest, and the sulfhydryl content in the EPI emulsion was the lowest.

### 3.2. Surface Hydrophobicity and Emulsifying Properties

Surface hydrophobicity is one of the key indicators used to influence protein surface behavior and emulsion properties. In an oil-in-water emulsion, the main driving force for the adsorption of protein at the interface is the hydrophobic interaction. The surface hydrophobicity of the protein depends on exposed non-polar groups. Generally, the more hydrophobic groups exposed, the better the surface hydrophobicity. As can be seen from Figure 2A, the surface hydrophobicity of pre-emulsification group is lower than that of the control. When the protein molecules are dispersed in the aqueous phase, most of the hydrophobic groups are located in the center of the molecules to reduce a link with the water phase and forms a hydrophobic area. Then, the hydrophobic groups are exposed and adsorbed on the surface of oil droplets after further emulsification [9], resulting in an increase in surface hydrophobicity in the control group whether in MP emulsion, EPI emulsion, or SPI emulsion (Figure 2A).

However, with the addition of NEM from 0 and 1 mM up to 5 mM, the surface hydrophobicity of the three kind protein emulsions all decreased first then increased. Based on the changes in sulfhydryl groups and disulfide bonds content, it can be speculated that, when disulfide bonds increased, the structure and hydrophilic surface of the protein molecule became stable. In contrast, the protein molecules unfolded and the embedded hydrophobic groups became exposed to the treated NEM, so the surface hydrophobicity changed [30]. Moreover, it also leads to a reduction in the solubility of the protein [31]. While 10 mM NEM was added, it was presumed that the protein molecular junction might be caused by a sharp decreased in disulfide bonds, which resulted in a decrease in surface hydrophobicity. However, the surface hydrophobicity of MP, EPI, and SPI as emulsifiers in controlling emulsion exhibited different results. The MP group has similar data to the SPI group but showed higher hydrophobicity than the EPI group. Maybe different proteins have diverse conformations. Hydrophobic amino acids buried in the core of globular protein should be exposed and adsorbed onto the surface of oil droplets, and the hydrophilic amino acids should be within the aqueous phase acting as a steric barrier against coalescence and flocculation [9].

Figure 2B shows that, in the pre-emulsification group or the control group, the EAI of MP emulsion was significantly higher than that of EPI emulsion or SPI emulsion since the EAI reflects the protein–protein and protein–lipid interactions. Amphipathic (hydrophilic and lipophilic) molecules in the myofibrillar protein structure are much more soluble compared with the other two proteins, which leads to strong interactions on the protein–lipid interface and protein–water interface [32]. The EAI of SPI emulsion was minimal because the close globular structure of SPI with poor molecular softness limited its foam ability and emulsifying properties [33,34] After adding NEM, the EAI of the three protein-type emulsion decreased with the increase in NEM content. It was presumed that the reduction in disulfide bond formation destroys the balance between protein hydrophilic molecules and lipophilic molecules [35]. An analysis of related data showed a significant positive correlation (*p* < 0.05) among the content of sulfhydryl and disulfide bonds and the activity of emulsification, which was similar to the results of Zhang and Lu [36].

ESI usually characterizes the rate of phase separation between aqueous and oil phases during emulsion storage [16]. Through adsorption, the protein can be adsorbed on the surface of oil droplets, thus forming a surface protein membrane to prevent the accumulation of oil droplets, which was effective at stabilizing the emulsion [8]. Table 1 shows that, at the same time, the ESI of the three types of emulsion is consistent with the change in EAI. With a longer duration time, the ESI value of each sample gradually worsened. In the same sample, with more NEM, the liquid and water phases separated faster. Consequently, the emulsion stability was worse. Similar results have been reported by Zhang and Lu, who found that, the higher the disulfide content, the easier it would increase the viscoelasticity of the interfacial film and the more stable the formed emulsion [36]. However, some studies have pointed out that a proper reduction in disulfide bonds can increase the flexibility of protein molecules and improve the emulsifying and foaming properties of the protein, thus further enhancing the stability of the foam of the emulsion [37]. Mcclements, Monahan, and Kinsella found that reactive SH groups were exposed when *β*-lactoglobulin in whey protein emulsion was absorbed at the oil–water interface [38]. Then, adsorbed SH groups formed disulfide bonds, resulting in continuous adsorption. Finally, protein molecules were constantly wrapped with droplets of molecules, thus forming a layer of viscoelastic interface protein membrane, making the emulsion more stable.

### 3.3. Comparison of Sulfhydryl Blocking Agent (NEM) on the Rheological Properties of Different Protein Emulsions

The rheological properties of proteins at the oil–water interface have become one of the important contents for evaluating the emulsifying properties of protein emulsions. The effect of a sulfhydryl blocking agent (NEM) on the rheological properties of different types of protein emulsions can be reflected by studying the changes in apparent viscosity under changes in the shear rate since the apparent viscosity of all samples was very close to 0 Pa s when the shear rate exceeded 1 s^−1^. In order to better distinguish the differences between the samples, only the interval with a shear rate of 0–1 s^−1^ was selected for research. As shown in Figure 3B, when 1 mM NEM was added to egg-white protein emulsion, the initial value of apparent viscosity was 2.17 Pa s. With the increase in shear rate, the apparent viscosity basically did not change with the change in shear rate, indicating that the emulsion fluid presents approximate rheological properties of Newtonian fluids. This phenomenon also exists in the 5 mM NEM group of soybean protein isolate emulsion [19]. Maneephan and Milena studied the rheological properties of unheated soybean protein isolate emulsion (10% soybean oil, *w*/*w*) and found that, when the protein concentration was 1.5% or 2.0%, the emulsion was close to the Newtonian fluid.

However, most samples showed that, the larger the shear rate, the smaller the apparent viscosity. It showed the phenomenon of shear thinning; thus, it could be concluded that most samples presented as pseudoplastic fluids. The reason for the shear thinning behavior was that, as the shear rate increased, the movement direction of the protein molecules gradually became uniform, frictional resistance decreased, and the chemical bonds such as hydrogen bonds broke, eventually leading to dissociation of the protein molecules [39,40,41]. Mutagenesis of SH groups and disulfide bonds could also alter the rheological properties of proteins. The apparent viscosity of the pre-emulsification group was lower than the control in all protein-emulsion groups, which may be due to the continued low-speed stirring emulsification, resulting in the increase in disulfide content in the protein emulsion, further stabilizing the emulsion structure. After adding NEM, the initial value of the apparent viscosity of myofibrillar protein emulsion was from 244 ± 5.13 to 118 ± 4.94, 365.5 ± 8.12, and 499.4 ± 7.05 Pa s. When the 1 mM NEM content was added in the emulsion, the initial apparent viscosity decreased, while with the increase in the amount of NEM added, the initial apparent viscosity increased significantly. This may be due to the decrease in the content of sulfhydryl, and some protein molecules in the emulsion undergo denaturation and aggregation, so the viscosity value increased. This phenomenon also occurs in egg-white protein emulsion and soybean protein isolate emulsion but with a subtle difference for different kinds of proteins used as an emulsifier.

### 3.4. Microstructure of Emulsion Droplets

As shown in Figure 4, the droplet sizes of pre-emulsification groups (Row 1) were larger than those of the control groups (Row 2). Hoogenkamp pointed out that, the smaller the particle size of the emulsified protein droplets, the more homogeneous the distribution and the more favorable the stability of the emulsions [42]. The control groups were formed after low-speed stirring for 30 min by pre-emulsion groups. This suggests that the emulsion droplets become smaller and more uniform through further low-speed stirring. It was consistent with the emulsifying properties (Table 1 and Figure 2B). However, when the control emulsion groups were treated with NEM, the emulsion droplet size significantly changed with different amounts of NEM. The droplet size with 1 mM NEM in emulsion was similar to that of the control, while the emulsion droplet size increased significantly when 5 mM NEM was added in the emulsion; 10 mM NEM added in the emulsion produced the maximum emulsion oil granules. This indicates that the reduction in disulfide bonds caused by NEM addition results in the aggregation of emulsion droplets. This suggested that the increase in disulfide bonds could contribute to the formation of a viscoelastic protein membrane and could attach to the surface of the oil droplets to stabilize and restrict the oil droplets, thus stabilizing the protein emulsion. This concept is depicted in Figure 5, where a schematic representation of protein emulsion droplets treated with or without NEM was provided. Disulfide interactions between protein molecules adsorbed at the interface are highlighted. Other molecular interactions (hydrogen bonds, ionic bonds, hydrophobic contacts, etc.), which may be present also, are not depicted [5]. Where NEM was used to block SH groups, less SS bonds were formed between the protein coating around the oil droplets, leading to the destabilization of emulsion droplets. Flocculation or especially coalescence induced increases in the droplet size. Figure 4 also shows that the spherical droplets in MP and SPI emulsions separated well, and the whole system was relatively consistent, which remained stable compared with EPI emulsion. Roesch and Corredig found that the droplet size could reach 1 µm in emulsions containing 1% SPI and 10% oil, but large particles were still widely distributed [43]. When the protein concentration increased to 2%, the droplet size decreased because of the reduction in interfacial tension, which was similar to the results shown in Figure 4. There was no significant difference in the changes about the three protein emulsions for the addition of 1 mM NEM compared with the control. In contrast, with NEM added, the content of sulfhydryl and disulfide bonds decreased, and the droplet sizes increased, accordingly. Damodara and Anand reported that whey protein emulsified with 1% whey protein (*w*/*w*) and 10% fat (*v*/*v*), when 20 mM NEM was added after 98 h, and formed smaller aggregates than those when 20 mM NEM was not added [3]. This indicates that the aggregation of intermolecular and interfacial proteins was caused by the intermolecular thiol-disulfide interconversion.

The droplets of the protein emulsion and the interaction between the droplets also affect the stability of the emulsion. The interaction between the droplets depends on the nature of the components (lipid type, protein type, and emulsifier), environmental factors pH, ionic strength, and temperature. The degree of change in droplet size followed MP < EPI < SPI. This may be due to the nature of the emulsion droplets caused by different protein types and the interaction between the protein droplet differences.

## 4. Conclusions

Sulfhydryl groups blocked by NEM and disulfide bond formation in the protein could affect the interaction between the protein molecules adsorbed on the surface of the oil droplets. Ultimately, it influences the interaction between the interfacial protein membrane and the aqueous phase, thus changing the physicochemical properties of the protein emulsion. The decrease in the content of sulfhydryl and disulfide bonds has a similar effect on different protein emulsions, which results in the deterioration of emulsifying activity and emulsifying stability. Furthermore, the emulsifying properties of myofibrillar proteins are significantly better than those of egg-white protein and soy protein isolate as the emulsifier. The microstructure shows that the emulsion droplet size increased with the content of NEM. The content of sulfhydryl groups and disulfide bonds decreased with the increase in NEM treatment level. This suggests that an increase in disulfide bonds could contribute to forming a better protein membrane and could attach to the surface of the oil droplets to stabilize protein emulsion. Certainly, the degree of influence is different. Different types of proteins with different structures and different contents of sulfhydryl and disulfide bonds lead to different emulsion properties.

## Figures and Tables

**Figure 1 foods-10-03079-f001:**
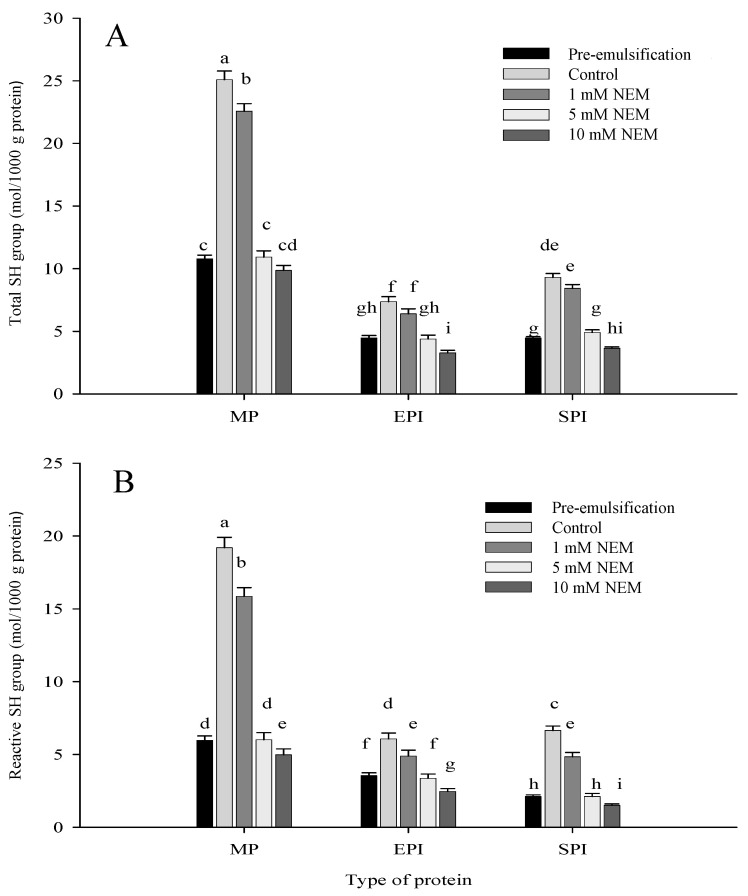
Effect of NEM treatments on the total sulfhydryl groups (**A**) and reactive sulfhydryl group (**B**) content of emulsions stabilized by different types of proteins. The lowercase letters indicate significant differences between means (*p* < 0.05). Pre-emulsification: emulsion obtained by high-speed homogenization at 12,000 rpm for 1 min without further treatment; control: emulsion with a subsequent moderate stirring process at 25 °C for 30 min but without NEM treatment; 1 mM NEM, 5 mM NEM, 10 mM NEM: emulsion treated with 1 mM, 5 mM, 10 mM NEM, respectively, followed by a moderate stirring process at 25 °C for 30 min. NEM: *N*-ethylmaleimide; MP: myofibrillar protein; EPI: egg-white protein isolate; and SPI: soybean protein isolate.

**Figure 2 foods-10-03079-f002:**
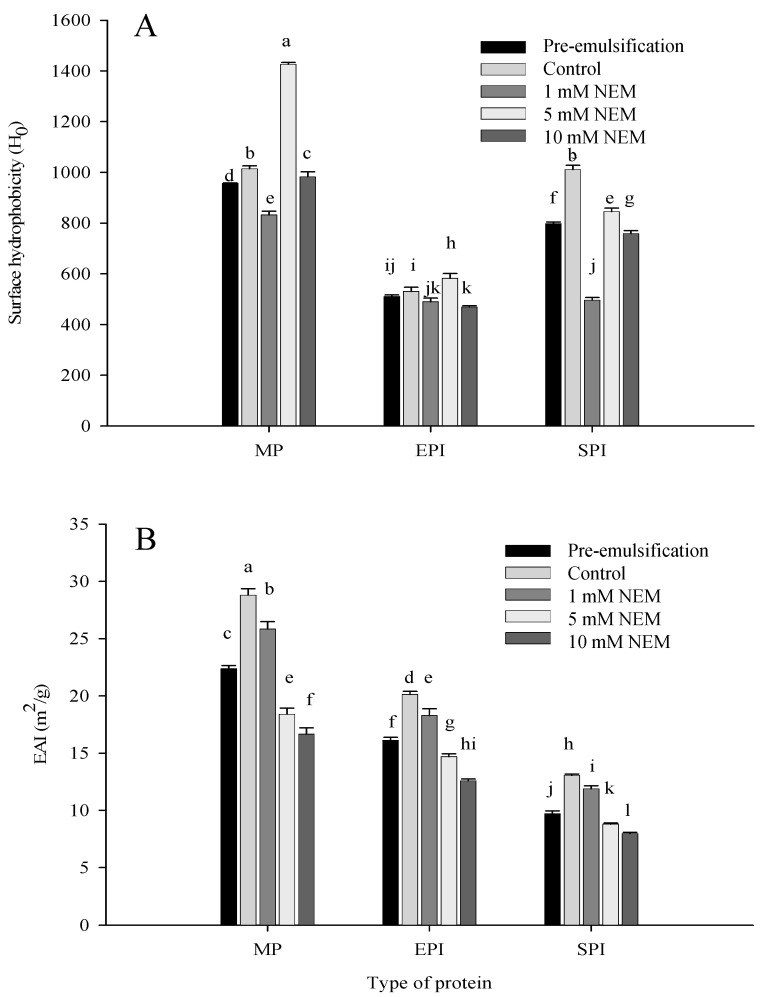
Changes in surface hydrophobicity (**A**) and emulsifying activity index (**B**) of the emulsions stabilized by different types of proteins (MP, EPI, and SPI) as affected by NEM (*N*-ethylmaleimide) treatment. The lowercase letters indicate significant differences between means (*p* < 0.05). Pre-emulsification: emulsion obtained by high-speed homogenization at 12,000 rpm for 1 min without further treatment; control: emulsion with a subsequent moderate stirring process at 25 °C for 30 min but without NEM treatment; 1 mM NEM, 5 mM NEM, and 10 mM NEM: emulsion treated with 1 mM, 5 mM, and 10 mM NEM, respectively, followed by a moderate stirring process at 25 °C for 30 min.

**Figure 3 foods-10-03079-f003:**
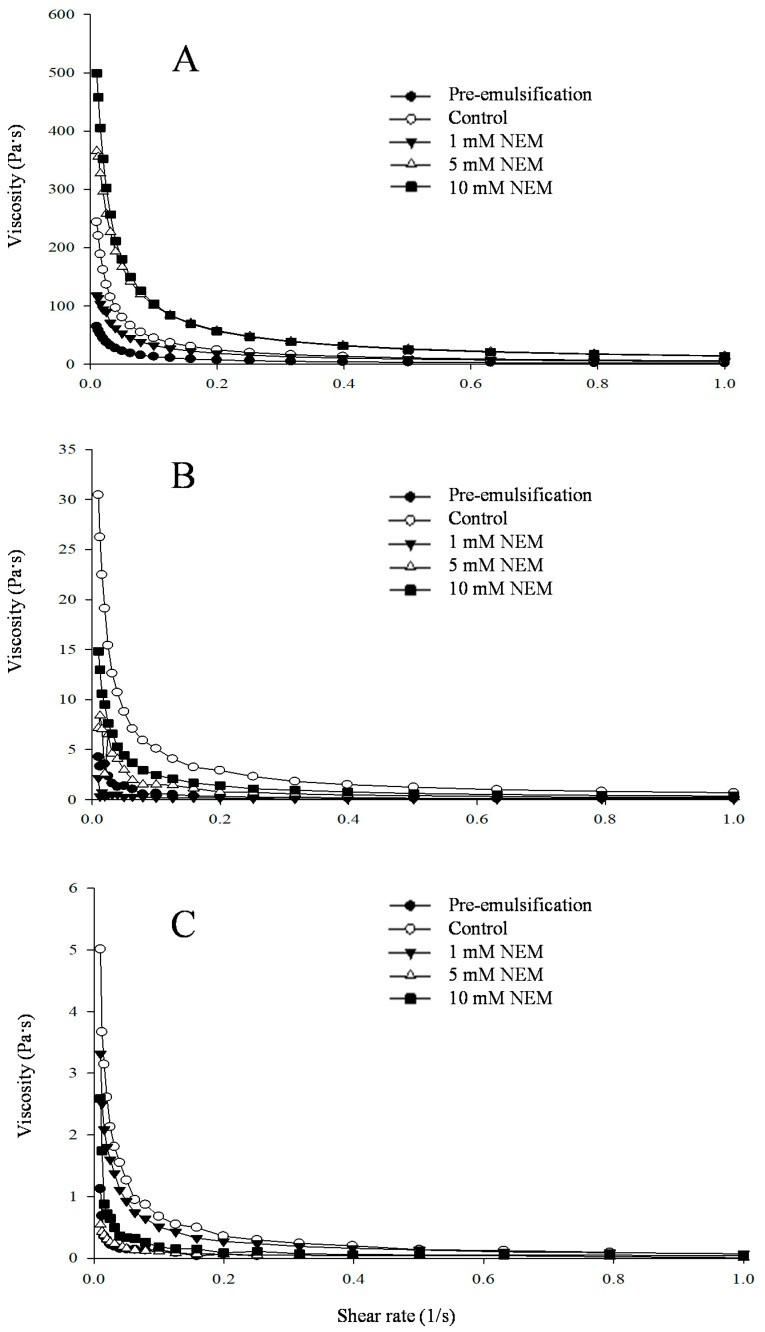
Changes in the apparent viscosity of emulsions stabilized by different types of proteins (MP, EPI, and SPI: myofibrillar protein, egg-white protein isolate, and soybean protein isolate) affected by NEM (*N*-ethylmaleimide) treatment: (**A**) MP, (**B**) EPI, and (**C**) SPI. Pre-emulsification: emulsion obtained by high-speed homogenization at 12,000 rpm for 1 min without further treatment; control: emulsion with a subsequent moderate stirring process at 25 °C for 30 min but without NEM treatment; 1 mM NEM, 5 mM NEM, and 10 mM NEM: emulsion treated with 1 mM, 5 mM, and 10 mM NEM, respectively, followed by a moderate stirring process at 25 °C for 30 min.

**Figure 4 foods-10-03079-f004:**
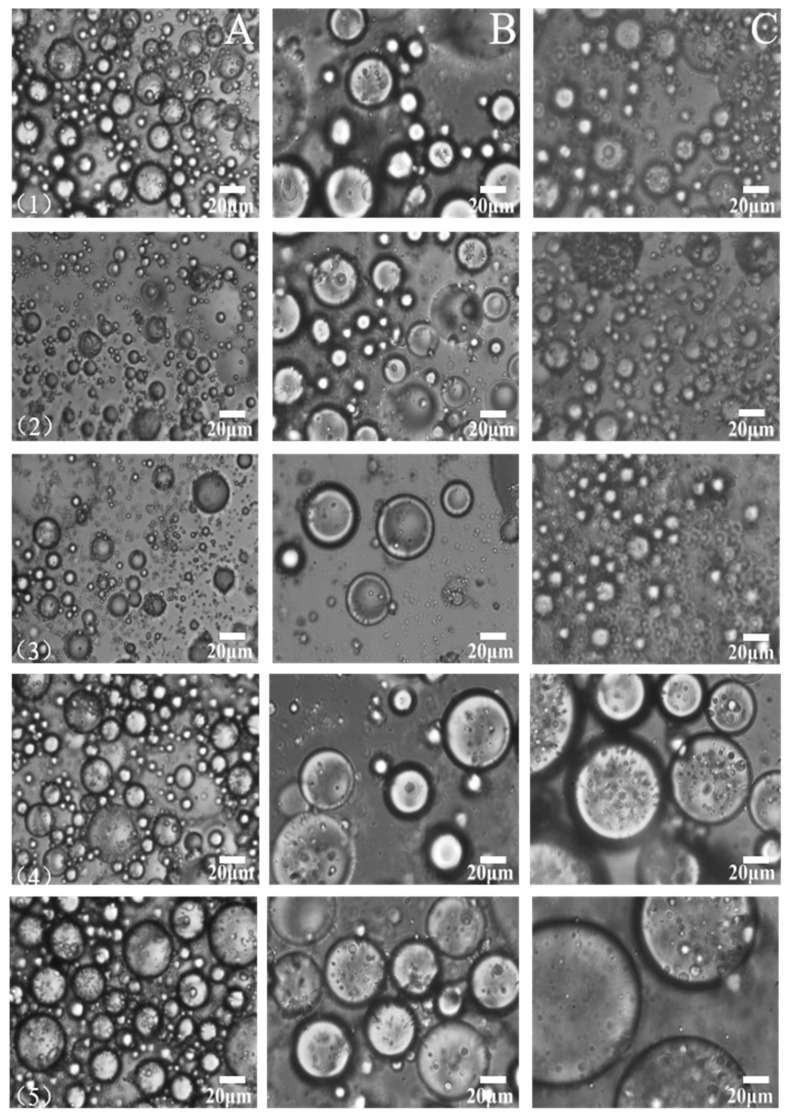
Effect of NEM (*N*-ethylmaleimide) content on the microstructure of emulsion stabilized by different types of protein (MP, EPI, and SPI: myofibrillar protein, egg-white protein isolate, and soybean protein isolate). Column A: MP as the emulsifier; column B: EPI as the emulsifier; column C: SPI as the emulsifier. Row (**1**), pre-emulsification group; row (**2**), control (continuing stirring for 30 min but no NEM added); row (**3**), 1 mM NEM (continuing stirring for 30 min and 1 mM NEM added); row (**4**), 5 mM NEM (continuing stirring for 30 min and 5 mM NEM added); and row (**5**), 10 mM NEM (continuing stirring for 30 min and 10 mM NEM added).

**Figure 5 foods-10-03079-f005:**
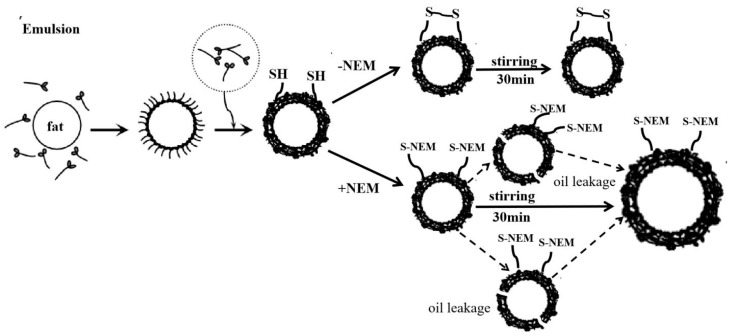
Schematic representation of the hypothetical mechanism of sulfhydryl blocked with or without NEM on the changes in emulsified lipid droplet size in protein emulsion. NEM: *N*-ethylmaleimide; SH: sulfhydryl; S-NEM: sulfhydryl blocked with *N*-ethylmaleimide.

**Table 1 foods-10-03079-t001:** The stability index of emulsions stabilized by different proteins (MP, EPI, and SPI) as affected by NEM treatment.

Treatment		Time (Min)				
		10 Min	30 Min	60 Min	120 Min	180 Min
MP	Pre-emulsification	0.95 ± 0.03 ^aA^	0.86 ± 0.01 ^bB^	0.77 ± 0.01 ^bcC^	0.70 ± 0.02 ^cD^	0.65 ± 0.01 ^cE^
	Control	0.98 ± 0.04 ^aA^	0.90 ± 0.02 ^aB^	0.84 ± 0.02 ^aC^	0.78 ± 0.03 ^aD^	0.714 ± 0.04 ^aE^
	1 mM NEM	0.96 ± 0.01 ^aA^	0.88 ± 0.03 ^abB^	0.80 ± 0.04 ^bC^	0.74 ± 0.02 ^bD^	0.67 ± 0.03 ^bE^
	5 mM NEM	0.90 ± 0.05 ^bA^	0.82 ± 0.01 ^cB^	0.74 ± 0.04 ^cdC^	0.61 ± 0.02 ^dD^	0.52 ± 0.03 ^dE^
	10 mM NEM	0.89 ± 0.07 ^bA^	0.80 ± 0.05 ^cB^	0.77 ± 0.05 ^dC^	0.66 ± 0.03 ^eD^	0.49 ± 0.03 ^eE^
EPI	Pre-emulsification	0.85 ± 0.01 ^bA^	0.66 ± 0.02 ^cB^	0.45 ± 0.04 ^bC^	0.42 ± 0.02 ^cD^	0.41 ± 0.04 ^cE^
	Control	0.91 ± 0.01 ^aA^	0.77 ± 0.02 ^aB^	0.64 ± 0.06 ^aC^	0.53 ± 0.03 ^aD^	0.49 ± 0.02 ^aE^
	1 mM NEM	0.89 ± 0.05 ^aA^	0.73 ± 0.07 ^bB^	0.62 ± 0.01 ^aC^	0.53 ± 0.02 ^bD^	0.46 ± 0.03 ^bE^
	5 mM NEM	0.81 ± 0.05 ^cA^	0.64 ± 0.05 ^cB^	0.41 ± 0.01 ^cC^	0.37 ± 0.01 ^cD^	0.32 ± 0.01 ^cE^
	10 mM NEM	0.78 ± 0.03 ^cA^	0.61 ± 0.04 ^dB^	0.39 ± 0.06 ^cC^	0.32 ± 0.01 ^dD^	0.27 ± 0.01 ^dE^
SPI	Pre-emulsification	0.74 ± 0.04 ^bA^	0.50 ± 0.05 ^bB^	0.45 ± 0.04 ^bC^	0.41 ± 0.01 ^bD^	0.40 ± 0.01 ^bE^
	Control	0.81 ± 0.01 ^aA^	0.70 ± 0.02 ^aB^	0.62 ± 0.01 ^aC^	0.56 ± 0.02 ^aD^	0.51 ± 0.01 ^aE^
	1 mM NEM	0.80 ± 0.08 ^aA^	0.68 ± 0.06 ^aB^	0.60 ± 0.02 ^aC^	0.55 ± 0.02 ^aD^	0.48 ± 0.01 ^aE^
	5 mM NEM	0.71 ± 0.03 ^bA^	0.47 ± 0.05 ^bB^	0.40 ± 0.01 ^cC^	0.37 ± 0.03 ^cD^	0.34 ± 0.01 ^cE^
	10 mM NEM	0.70 ± 0.03 ^bA^	0.44 ± 0.03 ^bB^	0.38 ± 0.01 ^cC^	0.35 ± 0.01 ^cD^	0.32 ± 0.01 ^cE^

Notes: ^a–e^ Different letters in the same column indicate significant differences of the same material between means at the same time (*p* < 0.05). ^A–E^ Different letters in the same row indicate significant differences of the same sample between means (*p* < 0.05). MP: myofibrillar protein; EPI: egg-white protein isolate; and SPI: soybean protein isolate.

## Data Availability

The datasets generated for this study are available on request to the corresponding author.

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
