# Peer review of "Role of Disulfide Bonds and Sulfhydryl Blocked by N-Ethylmaleimide on the Properties of Different Protein-Stabilized Emulsions"

_foods, 2021, doi:10.3390/foods10123079_

Round 1

Reviewer 1 Report

This manuscript describes an experimental investigation of the role of sulfhydryl groups and disulfide bonds in emulsions stabilized by food protein ingredients (myofibrillar protein, egg-white protein and soy protein). The authors have found that the addition of N-ethylmaleimide (NEM), which is well known to block intermolecular sulfhydryl groups, results in a reduction in disulfide bond formation, the destruction of the internal protein structure, and modification in the restriction of the protein membrane on the oil droplets. The authors’ claim for originality of this work is based on the statement in the last sentence of their introductory paragraph that the influence of sulfhydryl and sulfide bonds on the properties of emulsions “has not been deeply studied”. This latter statement is unfortunately not supported by a simple survey of the literature.

While the methodologies employed are quite routine, they are generally appropriate, and the results obtained do seem reliable and reasonably consistent. But the authors rely on the extremely outdated method of Pearce & Kinsella to determine emulsifying capacity and emulsion stability. The limitations of this method are now very well established in the literature. In modern research studies, it is recognized that more reliable emulsification data are to be derived from time-dependent light-scattering measurements of droplet-size distributions (see, for instance, the standard textbook on food emulsions by McClements).

The microscopy pictures in Fig. 4 indicate that the emulsions studied here are extremely coarse and polydisperse. This limits the likely significance of the reported stability and rheological data, including the reliability of the authors’ conclusion that that, although NEM-blocking of sulfhydryl groups gives a qualitatively similar effect for the three systems studied, there is a “subtle difference” in behaviour of the different food proteins. Overall, although the authors’ interpretations of their findings are largely plausible, they do not offer anything particularly new. The schematic diagram in Fig. 5 does not consider the likely important role of droplet flocculation as a precursor to droplet coalescence.

The manuscript is confusing to read because of the disorganized referencing presentation. The reference numbering in the text does not match the numbering in the reference list.

Authors are suggested to refer to the paper by Dickinson & Matsumura entitled “Time-dependent polymerization of β-lactoglobulin through disulfide bonds at the oil–water interface in emulsions” published in International Journal Biological Macromolecules (vol. 13, pp. 26-30). 

The Dickinson & Matsumura article describes time-dependent intermolecular sulfhydryl–disulfide interchange involving whey proteins adsorbed at the interface in oil-in-water emulsions at neutral pH. The 1991 paper provided the first direct experimental evidence that intermolecular disulfide bond formation occurs on adsorption of globular proteins at the oil–water interface. In addition, the results for the mixed whey protein system indicated that, in food emulsions containing a complex mixture of individual proteins, there is the possibility of forming hybrid polymers in the adsorbed layer (i.e. β-lactoglobulin linked to α-lactalbumin). The investigators found no polymerization of the adsorbed protein in emulsions prepared with pure α-lactalbumin after 72 h, or in emulsions made with β-lactoglobulin in the presence of NEM. Taken together with earlier time-dependent surface shear viscosity measurements (Dickinson et al., Int. J. Biol. Macromol., 1990, vol. 12, pp. 189-194), the study demonstrated the important role of free sulfhydryl groups in the development of the high surface viscoelasticity of adsorbed globular proteins at the oil–water interface.

Author Response

This manuscript describes an experimental investigation of the role of sulfhydryl groups and disulfide bonds in emulsions stabilized by food protein ingredients (myofibrillar protein, egg-white protein and soy protein). The authors have found that the addition of N-ethylmaleimide (NEM), which is well known to block intermolecular sulfhydryl groups, results in a reduction in disulfide bond formation, the destruction of the internal protein structure, and modification in the restriction of the protein membrane on the oil droplets. The authors’ claim for originality of this work is based on the statement in the last sentence of their introductory paragraph that the influence of sulfhydryl and sulfide bonds on the properties of emulsions “has not been deeply studied”. This latter statement is unfortunately not supported by a simple survey of the literature.

Answer: Thanks for your excellent suggestions. We have added the reference and deleted the misleading sentence. In the experiments, we found that emulsion oil droplets became bigger with NEM increase. it suggested disulfide bond in the protein membrane contributed to restrict the oil droplets. It seems a simple survey, but I think it can give another S-S band evidence to supported this interesting phenomenon.

While the methodologies employed are quite routine, they are generally appropriate, and the results obtained do seem reliable and reasonably consistent. But the authors rely on the extremely outdated method of Pearce & Kinsella to determine emulsifying capacity and emulsion stability. The limitations of this method are now very well established in the literature. In modern research studies, it is recognized that more reliable emulsification data are to be derived from time-dependent light-scattering measurements of droplet-size distributions (see, for instance, the standard textbook on food emulsions by McClements).

Answer: Some classic methods may seem to be outdated, but the results were reliable with enough replications. Certainly, thank you for your excellent suggestion, I should choose the standard methods as you suggested.

The microscopy pictures in Fig. 4 indicate that the emulsions studied here are extremely coarse and polydisperse. This limits the likely significance of the reported stability and rheological data, including the reliability of the authors’ conclusion that that, although NEM-blocking of sulfhydryl groups gives a qualitatively similar effect for the three systems studied, there is a “subtle difference” in behaviour of the different food proteins. Overall, although the authors’ interpretations of their findings are largely plausible, they do not offer anything particularly new. The schematic diagram in Fig. 5 does not consider the likely important role of droplet flocculation as a precursor to droplet coalescence.

Answer: When the control emulsion groups were treated with NEM, the emulsion droplets size occurred significantly changes with the different amount of NEM especially 5 mM and 10 mM added. It speculated that disulfide formation differed with NEM amount. In other words, flocculation or coalescence of emulsion droplets may be due to the formation of  SH-NEM and S-S bonds. So, I made a schematic representation.

The manuscript is confusing to read because of the disorganized referencing presentation. The reference numbering in the text does not match the numbering in the reference list.

Answer: Sorry for our terrible faults, we have organized all the references.

Authors are suggested to refer to the paper by Dickinson & Matsumura entitled “Time-dependent polymerization of β-lactoglobulin through disulfide bonds at the oil–water interface in emulsions” published in International Journal Biological Macromolecules (vol. 13, pp. 26-30). 

Answer: Excellent suggestions. We have referred this paper.

The Dickinson & Matsumura article describes time-dependent intermolecular sulfhydryl–disulfide interchange involving whey proteins adsorbed at the interface in oil-in-water emulsions at neutral pH. The 1991 paper provided the first direct experimental evidence that intermolecular disulfide bond formation occurs on adsorption of globular proteins at the oil–water interface. In addition, the results for the mixed whey protein system indicated that, in food emulsions containing a complex mixture of individual proteins, there is the possibility of forming hybrid polymers in the adsorbed layer (i.e. β-lactoglobulin linked to α-lactalbumin). The investigators found no polymerization of the adsorbed protein in emulsions prepared with pure α-lactalbumin after 72 h, or in emulsions made with β-lactoglobulin in the presence of NEM. Taken together with earlier time-dependent surface shear viscosity measurements (Dickinson et al., Int. J. Biol. Macromol., 1990, vol. 12, pp. 189-194), the study demonstrated the important role of free sulfhydryl groups in the development of the high surface viscoelasticity of adsorbed globular proteins at the oil–water interface.

Answer: Thanks for recommend this paper; it can help me much to understand the role of disulfide in protein membrane.

the results indicate the important role of free sulfhydryl groups in the development of the high surface viscoelasticity of adsorbed globular proteins at the oil-water interface

They give direct evidence that similar conformational changes and the associated intermolecular disulfide bond formation occurs on adsorption of b-lactoglobulin at the oil water interface.

Reviewer 2 Report

The article by Wu et al is focused on the study of emulsions stabilized by proteins of different origin (myofibrillar, egg-white, soybean) in order to investigate the role of SH groups and disulfide bonds. N-ethylmaleimide (NEM) was used at different concentration to block the sulfhydryl groups, reducing the disulfide bond formation.

Emulsifying properties, viscosity of emulsions as well as microstructure of the different system were reported.

There are some typo and grammatical errors, and check the corrispondence of the reference in the text.

abstract:

  • line 13: “The addition of NEM to block the sulfhydryl groups resulted in a reduction of the content of disulfide bonds formation”, I suggest to remove “of the content” or “formation”

introduction:

  • reference 4 is not correct; I think you are talking about ref 37
  • line 71 “emulsion emulsified” maybe you can change with “emulsion stabilized by”

paragraph 2.2

  • line 88. You are mentioning the Biuret method, please add a reference.

Paragraph 3.1

-line 186: You wrote “Reactive SH decreased much more with 10mM NEM treated than that treated with 5 mM NEM.” Looking at the Fig 1B it is possible to see a decrement but I would not say “much more”, considering the value with 5mM of NEM.

Line 267 Suggest to remove the dot between emulsion and since (one sentence)

Paragraph 3.3.

Why do you choose this range of shear rate?

As you report in method section and in the graph I suggest to report shear rate as 1/s instead of Hz in the text.

  • Line 305 “Since the apparent …. 1Hz” I suggest to move the sentence later in the text when presenting the results. Also, is this sentence true also for emulsion stabilized with MP with 10mM of NEM?

First you say that emulsions present approximate Newtonian behaviour and after you say that most sample were pseudoplastic. Could you be more clear about which sample show this behaviour.

-line 313 Missing the author of the reference you are introducing

-line 339 “This phenomenon also occurs in egg white protein emulsion and soybean protein isolate emulsion but with subtle difference.” From FIG 3C seems like the SPI has a different behaviour, whit an increase of viscosity with the addition of 1mM of NEM.

Have you considered also the z-potential value for the stability of the emulsions?

Were there also differences in the visual appearance of the emulsions when the NEM concentration was increased?

In your opinion could the amount of reactive SH be useful information in the choice of proteins to be used as stabilizer in the production of emulsion?

Author Response

The article by Wu et al is focused on the study of emulsions stabilized by proteins of different origin (myofibrillar, egg-white, soybean) in order to investigate the role of SH groups and disulfide bonds. N-ethylmaleimide (NEM) was used at different concentration to block the sulfhydryl groups, reducing the disulfide bond formation.

Emulsifying properties, viscosity of emulsions as well as microstructure of the different system were reported.

There are some typo and grammatical errors, and check the corrispondence of the reference in the text.

abstract:

  • line 13: “The addition of NEM to block the sulfhydryl groups resulted in a reduction of the content of disulfide bonds formation”, I suggest to remove “of the content” or “formation”

 Answer: Deleted as suggested, thanks.

introduction:

  • reference 4 is not correct; I think you are talking about ref 37

Answer: Thank you for your suggestions, we have checked all the references and corrected them.

  • line 71 “emulsion emulsified” maybe you can change with “emulsion stabilized by”

  Answer: Done as suggested, thanks.

paragraph 2.2

  • line 88. You are mentioning the Biuret method, please add a reference.

Answer: Done as suggested, thanks.

Paragraph 3.1

-line 186: You wrote “Reactive SH decreased much more with 10mM NEM treated than that treated with 5 mM NEM.” Looking at the Fig 1B it is possible to see a decrement but I would not say “much more”, considering the value with 5mM of NEM.

Answer: Sorry for misleading. “Reactive SH decreased much more with 5 mM NEM treated than that treated with 1 mM NEM”. In the text, we compared the effect of 5mM NEM and 1mM NEM on reactive SH. So, I corrected them.

Line 267 Suggest to remove the dot between emulsion and since (one sentence)

 Answer: Done as suggested, thanks.

Paragraph 3.3.

Why do you choose this range of shear rate?

Answer: The apparent viscosity of all samples was very close to 0 Pa·s when the shear rate exceeded 1 s-1. In order to better distinguish the differences between the samples, only the interval with a shear rate of 0-1 s-1 was selected for research.

As you report in method section and in the graph I suggest to report shear rate as 1/s instead of Hz in the text.

 Answer: Done as suggested, thanks.

  • Line 305 “Since the apparent …. 1Hz” I suggest to move the sentence later in the text when presenting the results. Also, is this sentence true also for emulsion stabilized with MP with 10mM of NEM?

 Answer: The sentence existed before the results; I think it is appropriate.

First you say that emulsions present approximate Newtonian behaviour and after you say that most sample were pseudoplastic. Could you be more clear about which sample show this behaviour.

Answer: A Newtonian fluid is a fluid in which the viscous stresses arising from its flow, at every point, are linearly correlated to the local strain rate the rate of change of its deformation over time. That is equivalent to saying those forces are proportional to the rates of change of the fluid's velocity vector as one moves away from the point in question in various directions. Pseudoplastic fluids are also referred to as shear-thinning fluids. The viscosity of these fluids will decrease with increasing shear rate. Typical examples for pseudoplastic fluids are polymer solutions and similar solutions of high molecular weight substances. At low shear rates, these liquids will experience the formation of shear stress. The shear stress results in the reordering of the molecules in order to reduce the overall stress.

-line 313 Missing the author of the reference you are introducing

  Answer: Sorry, “the author” has been added. Thanks

-line 339 “This phenomenon also occurs in egg white protein emulsion and soybean protein isolate emulsion but with subtle difference.” From FIG 3C seems like the SPI has a different behaviour, whit an increase of viscosity with the addition of 1mM of NEM.

 Answer: yes, it seems to have a different behavior..

Have you considered also the z-potential value for the stability of the emulsions?

Answer: Excellent suggestions, but we did not consider z-potential value for the emulsion stability.

Were there also differences in the visual appearance of the emulsions when the NEM concentration was increased?

Answer: yes, they had a little difference but not significantly.

In your opinion could the amount of reactive SH be useful information in the choice of proteins to be used as stabilizer in the production of emulsion?

Answer: yes, disulfide bonds will be formed by the amount reactive SH. Thanks.

Round 2

Reviewer 1 Report

In response to the reviewer's comments, the authors have moderated their claims for novelty and have included the important missing reference. They have also rectified the disorganization of the original reference list.